# Phytosynthesis of Silver Nanoparticles Using *Mansoa alliacea* (Lam.) A.H. Gentry (Bignoniaceae) Leaf Extract: Characterization and Their Biological Activities

**DOI:** 10.3390/pharmaceutics16101247

**Published:** 2024-09-25

**Authors:** Johana Zuñiga-Miranda, Saskya E. Carrera-Pacheco, Rebeca Gonzalez-Pastor, Arianna Mayorga-Ramos, Cristina Rodríguez-Pólit, Jorge Heredia-Moya, Karla Vizuete, Alexis Debut, Carlos Barba-Ostria, Elena Coyago-Cruz, Linda P. Guamán

**Affiliations:** 1Centro de Investigación Biomédica CENBIO, Facultad de Ciencias de la Salud Eugenio Espejo, Universidad UTE, Quito 170527, Ecuador; johana.zuniga@ute.edu.ec (J.Z.-M.); saskyacarrera@gmail.com (S.E.C.-P.); rebeca.gonzalez@ute.edu.ec (R.G.-P.); arianna.mayorga@ute.edu.ec (A.M.-R.); jorgeh.heredia@ute.edu.ec (J.H.-M.); 2Centro de Nanociencia y Nanotecnología, Universidad de Las Fuerzas Armadas ESPE, Sangolquí 171103, Ecuador; ksvizuete@espe.edu.ec (K.V.); apdebut@espe.edu.ec (A.D.); 3Departamento de Ciencias de la Vida y Agricultura, Universidad de las Fuerzas Armadas ESPE, Sangolquí 171103, Ecuador; 4Escuela de Medicina, Colegio de Ciencias de la Salud Quito, Universidad San Francisco de Quito (USFQ), Quito 170901, Ecuador; cbarbao@usfq.edu.ec; 5Instituto de Microbiología, Universidad San Francisco de Quito (USFQ), Quito 170901, Ecuador; 6Carrera de Ingeniería en Biotecnología de los Recursos Naturales, Universidad Politécnica Salesiana, Quito 170143, Ecuador; ecoyagoc@ups.edu.ec

**Keywords:** nanoparticles, silver nanoparticles, biological activity, green synthesis, sustainable approach, anti-tumor

## Abstract

**Background**. *Mansoa alliacea* is a native plant renowned for its medicinal properties in traditional healing in the Amazon Region. This plant is rich in polyphenols, flavonoids, anthocyanins, phenolic acids, tannins, ketones, triterpenes, as well as other bioactive compounds. **Objectives**. This study aims to develop an innovative, eco-friendly method for synthesizing silver nanoparticles using an aqueous extract of *M. alliacea* (Ma-AgNPs), enhancing the biological activities of AgNPs by leveraging the therapeutic potential of the plant’s bioactive compounds. **Methods**. Silver nanoparticles were synthesized using the aqueous extract of *M. alliacea*. The biological activities of Ma-AgNPs were assessed, including antibacterial, anti-inflammatory, antioxidant, antitumor, and anti-biofilm effects, along with evaluating their hemolytic activity. **Results**. Quantitative analysis revealed that Ma-AgNPs exhibit potent antibacterial activity against multidrug and non-multidrug-resistant bacteria, with MIC values ranging from 1.3 to 10.0 µg/mL. The Ma-AgNPs significantly reduced NO production by 86.9% at 4 µg/mL, indicating strong anti-inflammatory effects. They demonstrated robust antioxidant activity with an IC50 of 5.54 ± 1.48 µg/mL and minimal hemolytic activity, with no hemolysis observed up to 20 µg/mL and only 4.5% at 40 µg/mL. Their antitumor properties were notable, with IC50 values between 2.9 and 5.4 µg/mL across various cell lines, and they achieved over 50% biofilm inhibition at concentrations of 30–40 µg/mL. **Conclusions**. These findings underscore the potential of Ma-AgNPs for biomedical applications, particularly in developing new antimicrobial agents and bioactive coatings with reduced toxicity. This research highlights a sustainable approach that not only preserves but also amplifies the inherent biological activities of plant extracts, paving the way for innovative therapeutic solutions.

## 1. Introduction

Nanotechnology, especially the synthesis of silver nanoparticles (AgNPs), has witnessed a pivotal shift toward green synthesis methods, prioritizing biocompatibility and sustainability. Utilizing plant extracts for this purpose aligns with environmental safety standards and increases the cost efficiency [1,2]. Among the various natural resources employed in green synthesis, plant species with rich phytochemical profiles have emerged as particularly promising candidates, offering unique advantages in both nanoparticle formation and biological activity [3,4,5].

The distinct physicochemical properties of AgNPs and their versatile biological activities make them a focal point of research. AgNPs have demonstrated broad-spectrum antibacterial activity, effectively targeting Gram-positive and Gram-negative bacteria, thereby addressing the critical issue of antibiotic resistance [6,7]. Furthermore, their antibiofilm properties against medically significant bacterial strains underscore their potential in managing infections related to biofilms, which are notoriously difficult to treat [8,9]. Additionally, AgNPs exhibit antioxidant properties, as evidenced by their effective scavenging of DPPH radicals, suggesting a role in combating oxidative stress-related conditions [10]. Their low hemolytic activity at moderate concentrations further underscores their biocompatibility, a crucial consideration for biomedical applications [11]. Moreover, the antiproliferative effects of AgNPs against various cancer cell lines highlight their potential as novel agents in anticancer therapies, potentially offering synergistic effects when combined with existing treatments [12]. In summary, the superior properties of AgNPs make them an ideal focus over other types of nanoparticles, underscoring their relevance in biomedical research [13,14,15].

In this context, the plant *Mansoa alliacea*, known for its ethnobotanical significance and diverse pharmacological properties, represents an ideal candidate for the green synthesis of AgNPs. The ethnobotanical significance of *Mansoa alliacea* lies in its traditional medicinal uses among Amazonian communities. This plant is valued for its diverse pharmacological properties, including being used as an antiseptic, diuretic, analgesic, and antipyretic agent [16]. Additionally, it is employed in folk medicine for treating conditions like high blood pressure, atherosclerosis, arthritis, and rheumatism. The chemical composition of *M. alliacea* includes compounds like allicin, sulfides, sterols, iridoides, isothiocyanates, naphthoquinones, alkaloids, saponins, flavones, vitamins E and C, as well as minerals like chromium and selenium, contributing to its therapeutic potential [16,17,18].

Research into the *Mansoa alliacea* application in silver nanoparticle synthesis is sparse. However, studies on similar plant extracts shed light on the promising biomedical potential of AgNPs [19]. Notably, their efficacy in combating antibiotic-resistant bacteria and their antibiofilm properties against significant bacterial strains underscores the critical role AgNPs could play in addressing infectious diseases. The selection of *Mansoa alliacea* is driven by its underexplored potential in nanoparticle synthesis, aligning with a broader research trend towards utilizing natural plant processes for environmentally friendly nanoparticle production. This approach aligns with several Sustainable Development Goals (SDGs), notably SDG 3 (Good Health and Well-being), by potentially providing new treatments for infectious and chronic diseases, and SDG 12 (Responsible Consumption and Production) by promoting sustainable and environmentally friendly synthesis methods [20].

This study aims to explore the synthesis of AgNPs using the aqueous leaf extract of *M. alliacea* and further assess the biological activities regarding antitumoral, antioxidant, antibiofilm, antimicrobial, and hemolytic capabilities. By leveraging the plant’s phytochemical properties, this method promotes eco-friendly practices and opens new pathways for therapeutic applications.

## 2. Materials and Methods

### 2.1. Reagents and Standards

The analytical-grade reagents used in this study included methanol sourced from Fisher Scientific Inc. (Waltham, MA, USA). Additional reagents such as chloroform, hydrochloric acid, HPLC-grade methanol, chloramphenicol, tetracycline, carbenicillin, nourseothricine, 1% crystal violet aqueous solution, 2,2-diphenyl-1-picrylhydrazyl (DPPH), and HPLC-grade reagents like dimethyl sulfoxide (DMSO), lipopolysaccharides (LPS), Griess reagent, and thiazolyl blue tetrazolium bromide (MTT) were obtained from Sigma-Aldrich (St. Louis, MO, USA). Standards including caffeic acid, gallic acid, ferulic acid, syringic acid, o-coumaric acid, p-coumaric acid, shikimic acid, vanillic acid, 3-hydroxybenzoic acid, 2,5-dihydroxybenzoic acid, quercetin, kaempferol, chlorogenic acid, naringin, and ascorbic acid were also purchased from Sigma-Aldrich. Culture media such as Muller–Hinton Agar (MHA), Tryptic Soy Broth (TSB), and Brain Heart Infusion Broth (BHI) were procured from BD Difco—Fisher Scientific Inc. (Waltham, MA, USA). Phosphate-Buffered Saline (PBS) 10×, pH 7.4, was obtained from Invitrogen (Carlsbad, CA, USA).

### 2.2. Bacterial and Fungal Strains

The bacterial strains used in this study were obtained from the American Type Culture Collection (ATCC, Manassas, VA, USA). Gram-negative strains: *Salmonella enterica* ATCC 14028, *Escherichia coli* ATCC 25922, and *Pseudomonas aeruginosa* ATCC 27853. Gram-positive strains: *Staphylococcus aureus* ATCC 25923, *Enterococcus faecalis* ATCC 29212, and *Bacillus cereus* (environmental isolate). Multidrug-resistant strains of *Klebsiella pneumoniae*, *Escherichia coli*, and *Pseudomonas aeruginosa* were provided by the National Health Institute of Ecuador (INSPI). Fungi: *Candida tropicalis* ATCC 13803. All strains were maintained at −80 °C with 25% (*v*/*v*) glycerol supplementation.

### 2.3. Synthesis of Silver Nanoparticles Using Mansoa alliacea Leaf Extract

One hundred healthy and fresh *M. alliacea* (identification code: 4431, Herbario QUPS-Ecuador, Appendix A) leaves were purchased in February from the local market in the Amazon Region, Puyo city, Ecuador (1°29′3.671″ S, 77°59′39.172″ O). Aqueous extraction of *M. alliacea* leaves followed the methodology of Prasad (2016), with the following modifications [21]: fresh and healthy leaves of *M. alliacea* were washed three times with distilled water before drying in the shade at room temperature. The dried leaves were ground into a fine powder using an electric lab blender. The sieve N35 (500 μm) was used to sift the leaf powder obtained. For each 6 g of leaf powder, 100 mL of hot distilled water at 80 °C was added. The mixture was kept on constant stirring for 10 min. The resultant extract was filtered through Whatman filter paper No.1 and a filter 0.45 μm. The cooled solution was stored at −80 °C. The aqueous extract was then dried using a freeze-dryer, and the dry extract was weighed.

Aqueous leaf extract of *M. alliacea* (20 mL) was added to 30 mL of 10 mM silver nitrate solution. The reaction was carried out on a hot plate at 60 °C for 10 min with continuous stirring. A change in the color of the solution indicates the formation of Ma-AgNPs. After that, the suspension was centrifuged at 13,000× *g* for 10 min. The resulting sediment was washed three times with distilled water. The pure Ma-AgNPs were freeze-dried and stored at 4 °C for further characterization and biological studies.

### 2.4. Quantification of Phenols

Quantification of total polyphenols in the aqueous extract of *M. alliance* leaves followed the methodology of Johnson et al. [22]. The extract was subjected to the Folin–Ciocalteu assay with microplate reading using a Cytation 5 BioTek (Agilent Technologies, Santa Clara, CA, USA) and expressed as gallic acid equivalents (GAE) per 100 g of dry extract.

Quantification of individual phenolic compounds was performed according to Coyago et al. [23]. The aqueous extract (Section 2.2) was filtered through a 0.45 µm PVDF and transferred to a vial for quantification on an RRLC 1200 liquid chromatograph (Agilent Technologies, USA) coupled to a DAD-UV-VIS detector operating between 220 to 500 nm and a Zorbax Eclipse Plus C18 column (4.6 × 150 mm, 5 µm) (Agilent Technologies, USA). The mobile phase was set at 1 mL/min and considered a linear gradient of a 0.01% aqueous formic acid solution (solvent A) and acetonitrile (solvent B). Thus, 100% A, 0 min; 95% A + 5% B, 5 min; and 50% A + 50% B, 20 min, wash and re-equilibration, 22 min. In turn, 20 µL of the aqueous extract was injected in duplicate.

The quantification of phenolics was based on a calibration curve containing stock solutions of 1 mg/mL standards of individual compounds such as caffeic acid, chlorogenic acid, chrysin, *p*-coumaric acid, *m*-coumaric acid, *o*-coumaric acid, ferulic acid, gallic acid, *p*-hydroxybenzoic acid, 3-hydroxybenzoic acid, 2,5-dihydroxybenzoic acid, kaempferol, luteolin, naringin, quercetin, rutin, shikimic acid, syringic acid, and vanillic acid. These standards were prepared and quantified individually with injection volumes of 3, 5, 10, 15, and 20 µL.

Chromatograms were analyzed with OpenLab ChemStation software (Version 2.15.26) (Agilent Technologies, Santa Clara, CA, USA), and phenolic compounds were identified by comparison of retention times and UV-vis spectra at specific wavelengths of 280, 320, and 370 nm. Phenol concentration was expressed as milligrams per 100 g dry weight (mg/100 g DW).

### 2.5. Characterization of Ma-AgNPs

The UV-Vis was used to determine the formation of Ma-AgNPs using Cytation5 multi-mode detection system (BioTek). The hydrodynamic diameter of the particles was measured by dynamic light scattering (DLS) using an LB-550 Nanoparticle Size Analyzer (Horiba, Japan). Samples were characterized by X-ray diffraction (XRD) to evaluate the crystalline structure of the particles. The extract with nanoparticles was allowed to dry at room temperature on zero-background support. For this purpose, an Empyrean X-ray diffractometer (Malvern Panalytical, The Netherlands) equipped with a copper X-ray tube (Kα radiation, λ = 1.54056 Å) was used. The XRD data were collected in the 2Θ range from 5° to 90° with a scan rate of 0.01° at 45 kV and 40 mA. Highscore© software (Version 4.9a (4.9.1.29739), Malvern Panalytical B.V., Almelo, The Netherlands) was used for data interpretation linked with the PDF-2 database from the International Center for Diffraction Data. Energy-dispersive X-ray spectroscopy (EDS) was used to analyze the samples’ elemental composition. This analysis was carried out using a Mira 3 field emission Scanning electron microscope (SEM) (Tescan, Brno, Czech Republic) equipped with a Bruker X-Flash 6–30 detector with a resolution of 123 eV at Mn Kα. The microstructure of the powders was observed at 80 kV by transmission electron microscopy (TEM) using a Tecnai G20 Spirit Twin transmission electron microscope (FEI, The Netherlands) equipped with an Eagle 4 k HR camera.

### 2.6. Antibacterial Activity of Ma-AgNPs

#### 2.6.1. Well Diffusion Assay

The antibacterial activity of the biosynthesized Ma-AgNPs was determined using the antibiogram technique for initial screening. The bacterial inoculum was prepared in 0.9% saline solution and adjusted to 0.5 MacFarland standard. This suspension was spread uniformly on solidified Muller–Hinton Agar (MHA) using a sterile swab. A fixed volume of about 80 μL with different concentrations of the green-synthesized Ma-AgNPs (400, 200, and 100 μg/mL) was added to the wells, and Petri plates were incubated at 37 °C for 24 h. The inhibition zones obtained were measured in millimeters. The antibiotics listed in Table 1 served as the positive control for non-multidrug-resistant and multidrug-resistant bacteria. Additionally, silver nitrate (AgNO_3_) and Ma extract were used as a control.

#### 2.6.2. Determination of the Minimal Inhibitory Concentration (MIC)

The bacterial inoculum was prepared in the brain–heart infusion broth (BHI) to a final cell density of 5 × 10^5^ CFU/mL. Stock solution of the Ma-AgNPs was prepared by resuspending it in water at 2 mg/mL. Several antibiotics were used as controls for growth inhibition at the recommended working concentrations for the tested strains, as shown in Table 1, while the negative control consisted of the microorganism suspension alone (without Ma-AgNPs). Both BHI alone and supplemented with the compounds at different concentrations were used as blanks.

The sensitivity of Ma-AgNPs was assessed via the microdilution method [24] and according to the Clinical and Laboratory Standards Institute (CSLI) guidelines [25] with the following modifications: First, from the stock solution, the Ma-AgNPs were serially diluted in water, then 10 μL of each dilution was added to 190 μL of bacterial suspension (5 × 10^5^ CFU/mL) to a total volume of 200 μL. The range of the final concentrations of Ma-AgNPs was from 1.3 μg/mL to 100 μg/mL in each well. The plates were then incubated at 37 °C for 20 h with constant shaking at 300 cpm (double orbital setting), and the OD_600_ was monitored every 15 min in a Cytation5 multi-mode detection system (BioTek). The MIC was determined after tracking the bacterial growth over 20 h in samples exposed to the tested suspension at different concentrations. These assays were performed at least in triplicate.

The MIC was defined as the lowest concentration of the antibacterial agent, which completely inhibited the growth of the microorganism as determined by the optical density at 600 nm.

### 2.7. Biofilm Inhibition Activity

The following strains were selected for both their clinical importance and their ability to form biofilms, a trait specific to certain species. *S. aureus* ATCC 25923, *Listeria monocytogenes* ATCC 13932, *Enterococcus faecalis* ATCC 29212, and the fungal strains *Candida tropicalis* ATCC 13803 were grown for the evaluation of Biofilm Inhibition. All strains were cultured in TSB+G (tryptic soybean medium supplemented with 1% glucose) overnight.

The biofilm inhibition assay was performed as described by Merritt et al. (2005) with some modifications [26]. Briefly, overnight cultures were diluted 1:100 and aliquoted in volumes of 130 µL into 96-well polystyrene microtiter plates. A range of concentrations of Ma-AgNPs (40 µg/mL–5 µg/mL) was prepared, and 20 µL of each dilution was added to the microtiter plate wells. Plates were incubated under static conditions at 37 °C for 24 h.

The dilution stock solutions were prepared in distilled water. Concentrations were tested in technical triplicates during three independent biological replicates. The positive control of all strains also received 20 µL of distilled water and was used for data analysis.

After the 24 h incubation period, the medium was carefully removed with a micropipette, and the plate was washed twice with PBS 1× buffer (pH 7.2). The plate was dried inside a laboratory oven at 60 °C for 1 h. Subsequently, 150 µL of 0.1% crystal violet solution was added to each well for 20 min at room temperature. The staining solution was discarded, and wells were washed 3 times with PBS 1× buffer (pH 7.2). Finally, 200 µL of 98% ethanol was added to each well incubated for 30 min at room temperature, and absorbance was measured at 590 nm. The biofilm inhibition percentage was calculated using the following formula:$$Inhibitory\;rate\;(\%)=\left(\frac{{OD}_{590\;nm}(Positive\;control)-{OD}_{590\;nm}(Sample)\;}{{OD}_{590\;nm}(Positive\;control)}\right)*100$$

### 2.8. Antioxidant Activity by DPPH

For this purpose, the 2,2-diphenyl-1-picrylhydrazyl (DPPH) radical scavenging assay was carried out using the following samples: nanoparticles in extract (Ma-AgNPs), nanoparticles in water (AgNPs), and the extract alone (Ma-extract). The stock solutions for both the Ma-AgNPs and the AgNPs were prepared by dissolving them in water to a final concentration of 100 µg/mL. The Ma extract was adjusted to a final concentration of 204.3 µg/mL in water. Ascorbic acid was used as a standard (positive control) and the stock was prepared in water to a concentration of 50 µg/mL. Other controls included all the reaction reagents except for the samples. Background interferences from the solvents and reaction reagents were subtracted from the activities before calculating the radical scavenging capacity (A_solvent_ and A_DPPH_). Additionally, a sample blank was prepared with the solvent and the sample, and it was subtracted during the percentage DPPH scavenging calculation (A_sample blank_).

This assay was adapted from the literature [21] and performed in 96-well microtiter plates. Briefly, the stock solutions of the nanoparticles, the extract, and the standard were prepared as serial two-fold dilutions in methanol to a final volume of 100 µL. Then, 100 µL of a 0.2 mM methanol solution of DPPH radicals were added to 100 µL of the methanolic solutions. The mixture was incubated in the dark at room temperature for 40 min, and the absorbance was read at 515 nm using a Cytation 5 (BioTek) plate reader.

The following formula was used to calculate the *% DPPH scavenging activity:*$$\%\;DPPH\;scavenging=100\;*\;\left[\frac{\left(A_{sample+DPPH\;}-A_{sample\;blank\;}\right)}{\left(A_{DPPH\;}-A_{solvent\;}\right)}\right]$$

IC_50_ values were used to express the antioxidant activity of each compound and represent the concentration that could scavenge 50% of the DPPH free radical. These values were determined by using GraphPad Prism version 10.3.1 (GraphPad Software, Corp., San Diego, CA, USA). The results are given as a mean ± standard deviation (SD) of experiments completed in triplicate.

### 2.9. Cell Culture Experiments

#### 2.9.1. Cell Lines

MDA-MB-231 (human breast adenocarcinoma, ATCC No. HTB-26), SK-MEL-3 (human melanoma ATCC No. HTB-69), HCT116 (human breast adenocarcinoma, ATCC No. CCL-247), HT2659 (human colorectal adenocarcinoma, ATCC No. HTB-38), HeLa (human cervical carcinoma, ATCC No. CCL-2), and NIH3T3 (mouse NIH/Swiss embryo fibroblasts, ATCC No. CRL-1658) cells were obtained from ATCC. THJ29T (human thyroid carcinoma, Cat. No. T8254) was obtained from Applied Biological Materials Inc. (abm, Richmond, BC, Canada). These tumor and non-tumor cell lines were cultured in Dulbecco’s Modified Eagle’s medium/F-12 (DMEM/F-12) with L-glutamine (Sigma-Aldrich, St Louis, MO, USA). RAW 264.7 cells (mouse macrophages, ATCC No. TIB-71) were obtained from ATCC and were grown in Roswell Park Memorial Institute (RPMI) with L-glutamine (Corning, New York, NY, USA). All cells were cultured at 37 °C, 5% CO_2_, and supplemented with 10% Fetal Bovine Serum (FBS) (Eurobio, Les Ulis, France) and 1% penicillin and streptomycin (Thermo Fisher Scientific, Gibco, Miami, FL, USA).

#### 2.9.2. Anti-Inflammatory Assay

RAW264.7 cells were plated in 24-well plates (flat bottom, Corning) at a density of 5 × 10^5^ cells/well in 1 mL of medium and incubated for 24 h. The cells were pre-treated for 4 h with dexamethasone (0.5 µg/mL), Ma-AgNPs at concentrations ranging from 0.5 to 4 µg/mL, or Ma extract at concentrations ranging from 34 to 68 µg/mL, in a final volume of 500 µL per well. Then, the cells were stimulated with 1 µg/mL lipopolysaccharides (LPS) for 18 h. Cells stimulated only with LPS were used as positive control for inflammation (NO production), LPS + dexamethasone was used as the positive control for anti-inflammatory activity, and cells incubated only with media were used as negative control. After the treatment, supernatants were collected to assess nitric oxide (NO) production using the Griess reagent system as previously described [26]. Specifically, 50 µL of Griess reagent was added to 50 µL of each supernatant, and the mixture was incubated for 10 min at room temperature in the dark. Absorbance was measured at 540 nm using a Cytation5 multimode detection system (BioTek, Winooski, VT, USA). Culture medium was used as the blank and a stock solution of NO was included as a positive control to confirm the functionality of the Griess reagent. NO production was quantified relative to cells treated with LPS alone, which were set as 100% NO production.

#### 2.9.3. Anti-Tumor Activity Assay

To assess the impact of the compounds on cell proliferation, cells were seeded in 96-well plates (flat bottom, Corning) at a density of 1 × 10^4^ cells/well. Then, 100 µL of Ma-AgNPs were added to the wells at final concentrations ranging from 0.52 to 33 µg/mL and incubated with the cells for 72 h. Additionally, 100 µL of Ma extract was also incubated with the cells at concentrations ranging from 8.87 to 567.5 µg/mL. Cells only with media were used as negative control and cisplatin was used as positive control. After incubation, the MTT (thiazolyl blue tetrazolium bromide) assay was performed as previously described [27,28]. Briefly, 10 µL of the MTT solution (5 mg/mL) was added to each well, and the plate was incubated for 1–2 h. The media were removed and 50 µL of DMSO were added to each well to dissolve the formazan crystals. Agitation was performed for 5 min to ensure proper dissolution, and the absorbance was measured at 590 nm using a Cytation5 multi-mode detection system (Biotek). Each data point was obtained from quadruplicates, and the experiment was repeated four times. To determine the IC_50_ (concentration of compound inhibiting 50% of cell proliferation), dose–response curves were generated using untreated cells as the 100% cell proliferation control. This analysis was performed using GraphPad Prism version 9.5 software (GraphPad Software, Corp.).

### 2.10. Hemolytic Activity

The hemolytic activity of the Ma-AgNPs was measured according to a previously described protocol [29]. Ten milliliters of defibrinated sheep blood was subjected to three consecutive washes with 1× PBS. A 1% erythrocyte suspension in 1× PBS was prepared after the washing steps. The erythrocyte suspension was then combined 1:1 with Ma-AgNPs, positive (10% Triton X-100) or negative (1X PBS) hemolysis controls, in a 96-well polypropylene plate, and incubated at 37 °C for 1 h. After the incubation, the resulting mixture underwent centrifugation (5 min, 1700× *g*), and the resulting supernatant was carefully transferred to a transparent flat-bottom 96-well plate, allowing for the quantification of absorbance at 405 nm using a Cytation 5 plate reader. Each experiment was conducted with three technical replicates, and the entire protocol was replicated three times. For each sample, the hemolysis rate was calculated according to the formula:$$HR\left(\%\right)=\frac{{OD}_{test}-{OD}_{neg}}{{OD}_{pos}-{OD}_{neg}}*100$$

### 2.11. Statistical Analysis

All experiments were obtained in triplicate, and the data were calculated as mean ± standard deviation (SD). A one-way or two-way ANOVA with Tukey and Dunnett tests determined the significance of differences in means across groups using the GraphPad Prism 10.3.1 software (San Diego, CA, USA). The *p* values < 0.05 were considered statistically significant.

## 3. Results

### 3.1. Quantification of Phenols

The total polyphenol content of the aqueous extract of *M. alliance* (Ma) reported 49.6 ± 1.1 mg GAE/100 g of dry extract, while the chromatographic analysis showed the presence of flavonoids such as luteolin and isomers, *o*-coumaric acid, gallic acid, 4-hydroxybenzoic acid, syringic acid, and apigenin (Figure 1), with their concentrations shown in Table 2.

### 3.2. Synthesis and Characterization of Ma-AgNPs

Adding an aqueous leaf extract of *M. alliacea* to a colorless silver nitrate solution resulted in a color change from pale yellow to reddish-brown (see Appendix A), providing qualitative evidence for the formation of Ma-AgNPs.

The formation of Ma-AgNPs was further confirmed by the presence of a peak at 420 nm in the UV-visible spectroscopy (Figure 2), a characteristic surface plasmon resonance absorbance band indicative of the reduction of silver ions (Ag^+^) to metallic silver (Ag^0^).

#### 3.2.1. Dynamic Light Scattering Measurement

Dynamic light scattering (DLS) was used to determine the size of the nanoparticles. DLS analysis is a standard tool for determining the hydrodynamic size distribution and particle size of biosynthesized Ma-AgNPs. A typical DLS result was plotted in Figure 3 indicating that the average size diameter of the Ma-AgNPs was 47.5 nm ± 14.8 nm. Its polydispersity index (PDI) was 0.097, with a PDI lower than 0.1 which means that the sample is monodisperse [30].

#### 3.2.2. X-ray Diffraction (XRD) Analysis

The X-ray diffractogram is shown in Figure 4. The background from the original file was subtracted for better curve visualization, but the analysis was performed on rough data. The analysis was performed using Highscore© software (Version: 4.9a(4.9.1.29739), Malvern Panalytical B.V., Almelo, The Netherlands, 2021), and zerovalent silver was identified. No silver oxide was found, or if it exists, its concentration is less than the detection limit, confirming that the synthesis process predominantly produced Ag^0^, consistent with PDF card number 01-089-3722 (PDF-2 from ICCD database), which describes a cubic system in the Fm-3 m group space. The (110) and (200) planes were identified at 38.1° and 44.3°, respectively. The crystalline size was estimated from the Scherrer equation: 16.0 nm.

#### 3.2.3. Transmission Electron Microscopy (TEM) Analysis

The microphotographs from TEM determined the morphology and size distribution of the nanoparticles. In Figure 5, it can be observed that particles were not agglomerated and were quasi-spherical in shape. The organic part was also visible. The size distribution was analyzed manually using ImageJ© (Version 1.52p, National Institutes of Health, Bethesda MD, USA, 2019) from 1126 nanoparticles. Considering a normal distribution, the mean size value was 12.3 nm with a standard deviation of ±10.0 nm. The difference between the DLS and TEM mean size values corresponds to the size of the surface organic coating [31], which, in our case, is approximately 35 nm. This result is consistent with DLS and XRD values.

#### 3.2.4. Energy Dispersive Spectroscopy (EDS) Analysis

The EDS spectrum obtained from the selected area of the sample (Figure 6A) exhibits the presence of Ag, O, Na, Mg, P, S, K, and Ca, which are well-dispersed throughout the sample. The silver emission energy was observed at 2.98 keV, corresponding to the Lα1 peak (Figure 6B). The other elements identified are likely from the reducing and stabilizing agent (plant extract). Elemental mapping further confirms the presence of Ag in the sample (Figure 6C,D).

### 3.3. Antibacterial Activity

The antimicrobial activity of Ma-AgNPs was investigated on six strains (*B. cereus*, *S. enterica* ATCC 14028, *E. coli* ATCC 25922, *S. aureus* ATCC 25923, *P. aeruginosa* ATCC 27853, and *E. faecalis* ATCC 29212) and three MDR bacteria (*K. pneumoniae*, *E. coli*, and *P. aeruginosa)* using the agar well diffusion method and by determining the MIC. The well diffusion method demonstrated the magnitude of the susceptibility of the pathogenic microorganisms (Figure 7). The mean diameter of the inhibition zones (in millimeters) containing the Ma-AgNPs suspension at three different concentrations is presented in Table 3. Ma-AgNPs showed a higher inhibitory effect against *E. faecalis* with an inhibition zone of 35.0 ± 4.2 mm, 31.33 ± 4.5 mm for *S. aureus* ATCC 25923, and 25.67 ± 5.1 mm for *P. aeruginosa* ATCC 27853, in comparison with the standard antibiotic. The aqueous extract did not produce a diffuse ring for any microorganisms. The MIC values ranged between 1.3 μg/mL and 10.0 μg/mL. These results confirm the high antimicrobial potency of Ma-AgNPs. The lowest MIC of 1.3 μg/mL was observed against *S. enterica* ATCC 14028, *E. coli* ATCC 25922, and *S. aureus* ATCC 25923. The multidrug-resistant bacteria *K. pneumoniae* (5 μg/mL), *E. coli* (5 μg/mL), and *P. aeruginosa* (10 μg/mL) were more sensitive to Ma-AgNPs compared to the *M. alliacea* extract which showed no activity.

### 3.4. Biofilm Inhibition Activity

This study tested the biofilm inhibition effects of Ma-AgNPs (40–5 µg/mL) by crystal violet staining. The MBIC (minimum biofilm inhibitory concentration) to inhibit at least 50% of biofilm growth was used for the statistical analysis. The biofilm formation of two bacterial strains *S. aureus* ATCC 25923 and *L. monocytogenes* ATCC 13932 was significantly inhibited by nanoparticle treatment at 40 and 30 µg/mL Ma-AgNP concentrations (ANOVA test: *p*  =  0.05). The biofilm formation of the bacterial strain *E. faecalis* ATCC 29212 and the fungal strain *C. tropicalis* ATCC 13803 was also significantly impacted after treatment with 40 µg/mL Ma-AgNPs (Figure 8). The MBIC_50_ and inhibition rate percentages are presented in Appendix A.

### 3.5. Antioxidant Activity

The antioxidant activity of the nanoparticles was evaluated using the DPPH assay, as shown in Table 4. The Ma-AgNPs exhibited the lowest IC_50_ value (3.41 µg/mL), followed by the AgNPs (5.54 µg/mL). Although the radical scavenging activity of the ascorbic acid standard was at least four times more potent than that of the nanoparticles, this difference was not statistically significant. In contrast, the IC_50_ for the Ma-extract was 59.74 µg/mL, representing a significant difference compared to the Ma-AgNPs, AgNPs, and the standard (*p* < 0.0001). These results suggest that although the plant extract does not show an outstanding antioxidant potential, it may provide a more favorable environment for the nanoparticles, thereby enhancing their antioxidant capabilities or contributing to the overall radical scavenging activity.

### 3.6. Anti-Inflammatory Activity

To assess the anti-inflammatory activity of Ma-AgNPs, RAW264.7 cells were stimulated with LPS for 18 h following pretreatment with the nanoparticles or the plant extract. The levels of NO in the culture media were then determined using the Griess method. The concentrations used were selected based on the IC_20_ values (Appendix A) to avoid cell death, ensuring that cell viability remained above 95% for all controls and samples throughout the experiment.

As shown in Figure 9, LPS stimulation significantly increased NO production, confirming the establishment of the inflammatory model. NO levels in the absence of LPS stimulation were comparable to those of the negative control, which was incubated with cell media alone. Dexamethasone, used as a positive control, substantially inhibited LPS-induced NO production, reducing it to 40.5% of the control value (100% NO production). In contrast, pretreatment with 4 µg/mL of Ma-AgNPs for 4 h prior to LPS stimulation decreased NO production to 86.9% of the control, indicating a lower level of inhibition compared to dexamethasone. The Ma extract at 68 µg/mL exhibited minimal activity, reducing NO production to 92% of the stimulated control, similar to the effect of the lowest concentration of Ma-AgNPs.

### 3.7. Anti-Tumor Activity

The cell proliferation of various tumors and one non-tumor cell line was measured after 72 h of exposure to Ma-AgNPs (Appendix A). Dose–response curves showed growth inhibition in a dose-dependent manner (Appendix A) and were used to calculate the inhibitory concentration values (IC_50_). Table 5 presents the growth-inhibitory effects of Ma-AgNPs and Ma extract compared to cisplatin, a widely used platinum-based chemotherapeutic drug. The IC_50_ values for Ma-AgNPs ranged from 2.9 µg/mL to 5.4 µg/mL across different cell lines. In contrast, the Ma extract exhibited no significant inhibitory activity (NA) against the tested cell lines. Cisplatin showed IC_50_ values ranging from 2.3 µg/mL to 9.5 µg/mL.

The Therapeutic Index (TI) was calculated and is shown in Table 5. This value represents the ratio between the drug dose that causes a therapeutic effect (in tumor cells) and the dose that causes toxicity (in non-tumor cells). For Ma-AgNPs, the TI ranged from 1.0 to 1.9 across different cell lines, while cisplatin CDDP presented TI values ranging from 0.4 to 1.6, depending on the cell line.

### 3.8. Hemolytic Activity

Despite the interesting properties of silver nanoparticles, particularly in biomedicine, the evidence suggests they may exhibit cytotoxic effects on mammalian cells [32], including hemolytic activity. Hemolysis, the rupture of red blood cells leading to the release of hemoglobin, is a crucial indicator of cytotoxicity [33], especially for materials intended for therapeutic use. To determine whether Ma-AgNPs exhibit hemolytic activity, an assessment was conducted following a previously described protocol. The hemolytic activity was evaluated by comparing the effects of Ma-AgNPs at concentrations of 10, 20, and 40 µg/mL with a positive hemolysis control (10% Triton X-100, Fisher Scientific Inc. Waltham, MA, United States) and a negative control (1× PBS). The results (Table 6) demonstrated that Ma-AgNPs do not exhibit hemolytic activity at concentrations up to 40 µg/mL. Extremely low *p*-values in all comparisons indicated highly significant differences, confirming that the hemolysis caused by Ma-AgNPs was significantly lower than that of the control (Table 6).

## 4. Discussion

### 4.1. Synthesis and Characterization of Ma-AgNPs

In our research, the aqueous leaf extract of *M. alliacea* played a dual role as both a reducing and stabilizing agent in the synthesis of silver nanoparticles. This aligns with findings from HPLC and Folin–Ciocalteu tests, which identified the extract as a rich source of phenolic compounds and flavonoids, consistent with other studies on *Mansoa alliacea*’s phytoconstituents [34,35]. Notably, compounds such as gallic acid [36,37], 4-hydroxybenzoic acid 20 [38], syringic acid [39], cumaric acid [40], luteolin [41], and apigenin [42] have been documented for their reducing capabilities, which are crucial for the conversion of Ag^+^ ions to Ag^0^ and their subsequent integration into nanoparticle structures [43,44].

The significance of phenolic compounds, including flavonoids, in nanoparticle synthesis cannot be overstated. They act as reducing agents, facilitating the electron donation to metal ions to form nanoparticles, and also serve as capping agents enhancing nanoparticle stability by preventing aggregation [21,45,46,47,48,49]. Additionally, the presence of proteins in the plant extract contributes to the synthesis process by trapping metal ions on their surface, thereby forming the nuclei that aggregate into nanoparticles [50].

An observable color transition in the Ma-AgNPs solution from pale yellow to reddish-brown within 10 min of reaction commencement qualitatively verified nanoparticle formation. This rapid color change, attributed to Surface Plasmon Resonance (SPR), demonstrates the efficiency and speed of the biological synthesis methods compared to other approaches, which often take 30 min or more [51,52]. SPR occurs when electromagnetic radiation interacts with the nanoparticles’ surface, resulting in the observed color shift [53].

UV-visible spectroscopy further corroborated the formation of Ma-AgNPs, with an absorbance peak around 420 nm, indicative of nanoparticle presence in the solution. This is consistent with absorption spectra from other studies, such as *Scabiosa atropurpurea* at 423 nm [54], *Oxalis stricta* around 450 nm [55], and *Lantana camara*, which exhibited a range of 420–450 nm, with particle sizes between 3.2–12 nm [56].

### 4.2. Antimicrobial Activity

Our study revealed that silver nanoparticles synthesized using *M. alliacea* leaf extract exhibited remarkable antimicrobial efficacy. The inhibition zones (IZ) for various bacteria strains surpassed those of standard antibiotics, with *E. coli* showing an IZ of 15.3 mm, *B. cereus* 19.0 mm, *S. enterica* ATCC 14028 17.3 mm, *S. aureus* ATCC 25923 25.9 mm, *P. aeruginosa* ATCC 27853 21.0 mm, and *E. faecalis* ATCC 29212 19.5 mm at lower concentrations used in the well diffusion test. This enhanced activity of Ma-AgNPs over other biogenically synthesized nanoparticles is notable [57]; for instance, nanoparticles synthesized with *Carduus crispus* extract demonstrated a 6 mm IZ against *E. coli* [58], while those obtained from *Gmelina arborea* extract exhibited IZs of 10 mm, 11.33 mm, and 11.66 mm against *B. cereus*, *P. aeruginosa*, and *S. aureus,* respectively [59]. Moreover, the biosynthesized silver nanoparticles in our study showed low MIC values for these bacterial strains, ranging from 1.3 to 5 μg/mL. This is significantly more potent than the ranges reported by other researchers, which are typically between 6.25 to 50 μg/mL [10,60,61].

Against multidrug-resistant bacteria, Ma-AgNPs at various concentrations outperformed traditional antibiotics, a finding consistent with nanoparticles synthesized from *Catharanthus roseus* and *Azadirachta indica*, which also showed high antibacterial efficacy [62]. Additionally, the MIC obtained in our study against multiresistant bacteria was in the range of 5–10 μg/mL, which is more effective than AgNPs synthesized using *Z. officinalis, M. piperita,* and *T. vulgaris,* whose ranges vary between 35 and 141 μg/mL [63]. AgNPs synthesized using *R. discolor* leaf extract demonstrated MIC values ranging from 0.93 to 3.75 mg/mL against MDR strains [64]. AgNPs synthesized using *Citrus maxima* peel extract demonstrated potent antibacterial activity against methicillin-resistant *S. aureus* with a MIC value of 8.27 µg/mL, which is consistent with our findings [65].

While not fully understood, the mechanism behind the antibacterial properties of AgNPs is believed to involve several actions, including enzyme inactivation, reactive oxygen species production, and the release of Ag^+^ ions that bind to and denature proteins [66,67,68]. Contrary to studies suggesting a higher efficacy of AgNPs against Gram-negative bacteria due to differences in the cell wall composition [69], our findings did not observe this trend, indicating that the antibacterial effectiveness of AgNPs synthesized from *M. alliacea* is robust across both Gram-positive and Gram-negative strains. This could be attributed to factors such as nanoparticle size, dispersion, and shape, underscoring the potential of Ma-AgNPs as versatile antimicrobial agents [54]. The synthesized Ma-AgNPs were equally effective against both antibiotic-susceptible and resistant strains. Lara et al. (2010) reported that AgNPs had antibacterial properties regardless of antibiotic resistance and this feature could be attributed to the bactericidal effect rather than bacteriostatic mechanisms [70].

### 4.3. Biofilm Inhibition Activity

Biofilms pose significant health challenges, especially when formed on medical devices like urinary catheters, leading to persistent infections. They are notoriously resistant to antimicrobial agents compared to planktonic cells, prompting extensive research into effective biofilm-control strategies [71]. Studies have consistently shown that both chemically synthesized and green-synthesized AgNPs are potent biofilm inhibitors, with green-synthesized AgNPs demonstrating enhanced activity against medically significant bacterial strains, including *S. aureus* and its multidrug-resistant variants, compared to their chemically synthesized counterparts [72,73,74].

In our research, the biofilm inhibitory effects of Ma-AgNPs (ranging from 5 to 40 µg/mL) and the *M. alliacea* extract were evaluated using crystal violet staining to determine the MBIC. Significant biofilm inhibition was observed in both fungal and bacterial strains at the highest concentration of Ma-AgNPs (40 µg/mL), with *L. monocytogenes* showing an 85% inhibition rate at 30 µg/mL. This rate of biofilm inhibition is notably high compared to other green-synthesized AgNPs, such as those derived from *Terminalia catappa*, which reported 45% inhibition at 100 µg/mL for *L. monocytogenes* [75]. The literature also describes antibiofilm MIC values for AgNPs from plant extracts like *Rhododendron ponticum, Mimusops elengi*, and *Pimpinella anisum* L. ranging significantly higher, from around 100 µg/mL to 300 µg/mL in *E. faecalis* biofilms [76,77,78]. Conversely, studies have documented lower MBIC values for green-synthesized AgNPs (1.28–8 µg/mL) in *S. aureus* biofilms [73,79], and AgNPs from *Dodonaea viscosa* and *Hyptis suoveolens* recorded a 10 µg/mL MBIC against *C. tropicalis* [80]. It is noteworthy that variations in MBIC outcomes could stem from differences in the methodology, such as the use of the crystal violet assay and the solvent type for extract preparation (aqueous vs. ethanolic), which might account for the discrepancies with our aqueous-based synthesis findings.

Overall, our study underscores the significant antibiofilm potential of AgNPs synthesized from *M. alliacea*, aligning with the broader scientific consensus on the efficacy of green-synthesized AgNPs. This evidence supports the broader application of *M. alliacea*-derived AgNPs in combating biofilm-related infections, providing a promising avenue for future antimicrobial strategies.

### 4.4. Antioxidant Activity

As reported in the results, the Ma-AgNPs do not show a statistically significant difference with the ascorbic acid standard. Therefore, the IC_50_ values derived from our experiments underscore the exceptional antioxidant capabilities of Ma-AgNPs, particularly evident in their DPPH radical scavenging efficacy. To contextualize these findings, we compared IC_50_ values reported for AgNPs synthesized from various plant extracts. For instance, AgNPs from the aqueous leaf extract of *Erythrina suberosa* (Roxb.) showed an IC_50_ value of 30.04 μg/mL [81], those from silky hairs of corn (*Zea mays* L.) recorded 385.87 µg/mL [82], from *Prosopis farcta* fruit extract had 700 µg/mL [83], and from ginger rhizome extract (*Zingiber officinale*) were observed at 68 µg/mL [84]. The superior antioxidant activity observed in AgNPs synthesized from *M. alliacea* (3.41–5.54 µg/mL, this study) is attributed to the rich bioactive constituents of the plant extract, including phenolic compounds and flavonoids known for their robust antioxidant properties. These compounds enhance the antioxidant potential of the resultant AgNPs when involved in their synthesis [85,86].

The IC_50_ value from the Ma extract (59.74 µg/mL) showed antioxidant potential; however, it was not comparable to the synthesized nanoparticles, as suggested by the statistical analysis. Nevertheless, as mentioned earlier, the extract may boost the antioxidant potential of the synthesized AgNPs.

Nanoparticles possessing potent antioxidant abilities are crucial for mitigating oxidative-stress-related ailments and protecting against the cellular damage induced by free radicals [87,88]. Moreover, they are efficient scavengers of reactive oxygen species (ROS), safeguarding cells from oxidative harm [89]. The diversity in IC_50_ values across studies underscores the variable antioxidant efficacies of AgNPs derived from different plant sources. This variability signals further research to decipher the mechanisms and refine the synthesis processes to maximize the antioxidant activity.

### 4.5. Anti-Inflammatory Efficacy

Assessing anti-inflammatory activity is crucial to understanding the potential therapeutic effects of a compound. AgNPs have been documented to exhibit anti-inflammatory activity by inhibiting the activation of immune cells and reducing the production of pro-inflammatory molecules like nitric oxide synthase (iNOS) and cyclooxygenase (COX) [85,90]. Similarly, *M. alliacea* is believed to hold anti-inflammatory effects due to its bioactive contents that directly modulate inflammatory pathways and neutralize harmful free radicals associated with oxidative damage [91,92,93]. In this context, the impact of dexamethasone (used as the positive control), Ma-AgNPs, and Ma extract on inhibiting NO production following LPS stimulation was investigated. Pre-treatment with 4 µg/mL of Ma-AgNPs for 4 h before LPS stimulation reduced NO production to 86.9% compared to the LPS control; however, this inhibition percentage was lower than that achieved by dexamethasone. Previous studies have reported a higher level of anti-inflammatory activity of AgNPs synthesized by green approaches, reducing NO production by 35–70% [94,95,96,97]. Most of these studies fail to compare the effect of the extract alone to the composites. Differences in the size and shape of the resulting Ma-AgNPs could be related to their reduced anti-inflammatory activity. Additionally, the effect of the extract used for synthesizing the nanoparticles needs to be considered [98]. These variations impact the physicochemical properties of Ma-AgNPs, their interactions with biological systems, and their anti-inflammatory activity. Further research is required to elucidate the extent of the *Mansoa alliacea* extract and its influence on the AgNPs on the inflammatory response [99].

### 4.6. Anti-Tumor Activity

Over the past few decades, there has been a growing interest in harnessing nanomaterials for cancer treatment. This has led to the development of innovative nanotechnology alternatives against various types of cancer that have shown promising therapeutic potential [100]. According to some studies, silver nanoparticles may exhibit a synergistic interaction with anticancer medications, enabling the administration of reduced dosages [101]. AgNPs can induce cytotoxic effects on cancer cells by modifying their structure, diminishing their viability, and triggering oxidative stress in diverse cancer cell types [102]. Ma-AgNPs exhibited notable efficacy in inhibiting HeLa cells (IC_50_: 2.9 µg/mL) and SK-MEL-103 cells (IC_50_: 4.4 µg/mL), indicating promising anticancer potential against these specific cancer types. However, higher IC_50_ values were observed for MDA-MB-231, HCT116, and HT29 cells, similar to the IC_50_ for NIH3T3 (5.4 µg/mL), a non-tumor cell line. Comparable or higher values have been reported for other AgNPs synthesized via green methods. These studies employed extracts of plants [103,104,105,106] and marine algae [107,108,109] to cap AgNPs, with IC_50_ values ranging between 2.31 and 132 µg/mL in various tumor cell lines. This suggests that Ma-AgNPs could offer a viable alternative for cancer treatment, particularly against specific cancer types.

Compared to CDDP, Ma-AgNPs exhibited similar or lower IC_50_ values in certain cases. Notably, CDDP demonstrated less efficiency in reducing proliferation in MDA-MB-231, SKMEL-103, and HCT116 cells (IC_50_ 9.5 µg/mL, 5.6 µg/mL, and 7.5 µg/mL, respectively). Interestingly, CDDP inhibited cell proliferation in NIH3T3 cells at a lower concentration (IC_50_ 3.7 µg/mL) than Ma-AgNPs, suggesting that CDDP might be more effective in limiting cell proliferation in non-tumor cells. Overall, Ma-AgNPs demonstrate consistently higher TI values than CDDP. The TI measures a compound’s selectivity between tumor and non-tumor cells, suggesting that Ma-AgNPs might have a stronger tendency to target cancerous cells over healthy ones, a desirable trait for an effective therapeutic agent. However, it is crucial to emphasize that these findings are preliminary. Further comprehensive studies, including preclinical and clinical trials, are essential to thoroughly evaluate the compound’s safety and effectiveness thoroughly, and its suitability as an anti-tumor treatment.

### 4.7. Hemolytic Assay

The hemolytic assay results offer crucial insights into the biocompatibility of Ma-AgNPs, a key consideration for their biomedical applications. At 10 µg/mL and 20 µg/mL concentrations, Ma-AgNPs exhibited negligible hemolytic activity [110], with hemolysis rates of 0% and 0.5%, respectively. These findings are notable, indicating that in a similar way to other AgNPs, Ma-AgNPs maintain the integrity of red blood cells even at moderately high concentrations [111]. A mild increase in hemolytic activity to 4.5% at a 40 µg/mL concentration suggests a dose-dependent relationship. Yet, this level remains significantly lower than the positive control (100% hemolysis induced by 10% Triton X-100), affirming the relative safety of Ma-AgNPs up to this concentration. However, the observed increment at higher concentrations highlights the necessity of precise dosage selection in biomedical applications to optimize efficacy and safety [112].

These results enhance the appeal of Ma-AgNPs as a biocompatible option for applications like drug delivery, wound healing, and anti-cancer therapies. The minimal hemolytic activity observed, even at elevated concentrations, not only emphasizes their potential safety in biomedical uses, but also mirrors the favorable biocompatibility trends associated with green-synthesized nanoparticles. This trend, evident in our findings and supported by the literature [113,114,115,116], suggests that biologically synthesized nanoparticles generally exhibit benign interactions with human tissues, especially red blood cells.

Our investigation into the hemolytic behavior of Ma-AgNPs adds to the growing understanding of nanoparticle biocompatibility, underscoring the importance of synthesis techniques and nanoparticle characteristics in influencing biological responses. While the results are encouraging, they underscore the imperative for ongoing research into dose-dependent effects and long-term safety to fully leverage the therapeutic capabilities of such nanoparticles responsibly. As nanomedicine evolves, studies like ours are essential in ensuring the safe and effective use of nanomaterials in clinical settings, with a focus on biocompatibility and minimal cytotoxicity.

## 5. Conclusions

This study demonstrates the green synthesis of silver nanoparticles (Ma-AgNPs) figure Susing *M. alliacea* aqueous leaf extract, leveraging its bioactive phenolic compounds and flavonoids to reduce Ag^+^ ions to Ag^0^. Compounds such as gallic acid, 4-hydroxybenzoic acid, syringic acid, coumaric acid, luteolin, and apigenin were instrumental in both nanoparticle formation and stabilization. The rapid color change due to Surface Plasmon Resonance (SPR) indicates an efficient synthesis process, confirmed by UV-visible spectroscopy showing an absorbance peak at around 420 nm, consistent with other plant-based nanoparticles.

Ma-AgNPs exhibited potent biological activities, including strong antimicrobial effects against non-multidrug-resistant and multidrug-resistant bacteria, significant biofilm inhibition, and antioxidant properties, as evidenced by the DPPH assay. Although anti-inflammatory effects were modest, Ma-AgNPs showed promise in regulating inflammatory responses. The nanoparticles demonstrated notable anticancer activity against cervical carcinoma and melanoma, with lower IC_50_ values and higher therapeutic indices (TI) than conventional drugs like CDDP, indicating enhanced selectivity for cancer cells.

The hemolytic assay results confirmed the biocompatibility of Ma-AgNPs, highlighting their potential for safe biomedical applications. These findings underscore the environmental benefits of green synthesis using *M. alliacea* and the potential of the synthesized nanoparticles as multifunctional agents for biomedical use.

Overall, this study advances green nanotechnology by utilizing natural resources to create nanoparticles with therapeutic potential and calls for further research into the underlying mechanisms and expanded biomedical applications.

## Figures and Tables

**Figure 1 pharmaceutics-16-01247-f001:**
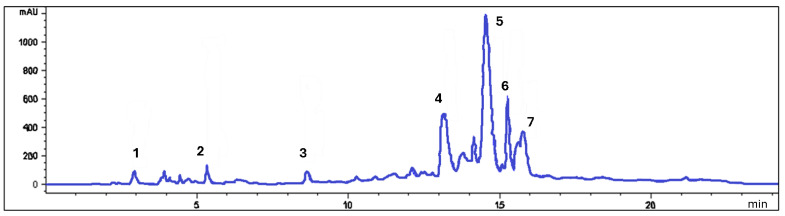
Chromatogram of *Mansoa alliace* phenolics at 280 nm. 1. Gallic acid; 2. 4-hydroxybenzoic acid; 3. siringic acid; 4. *o*-cumaric acid; 5. luteolin; 6. luteolin isomer; 7. apigenin.

**Figure 2 pharmaceutics-16-01247-f002:**
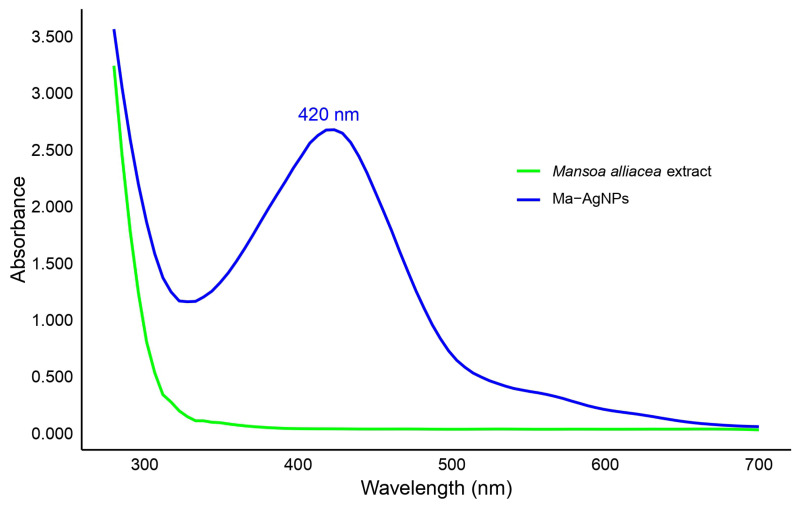
UV-Vis spectrum of the nanoparticle dispersion.

**Figure 3 pharmaceutics-16-01247-f003:**
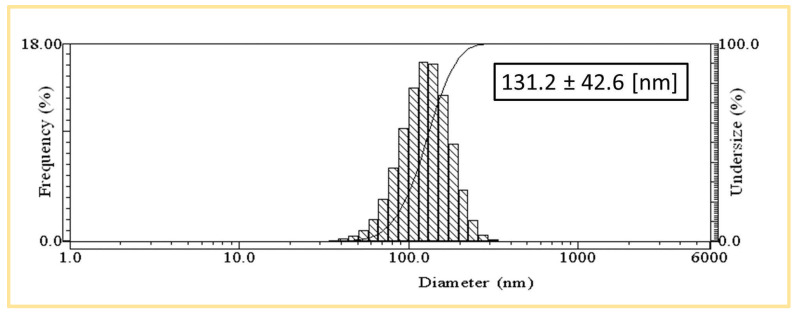
Dynamic light scattering analysis of Ma-AgNPs.

**Figure 4 pharmaceutics-16-01247-f004:**
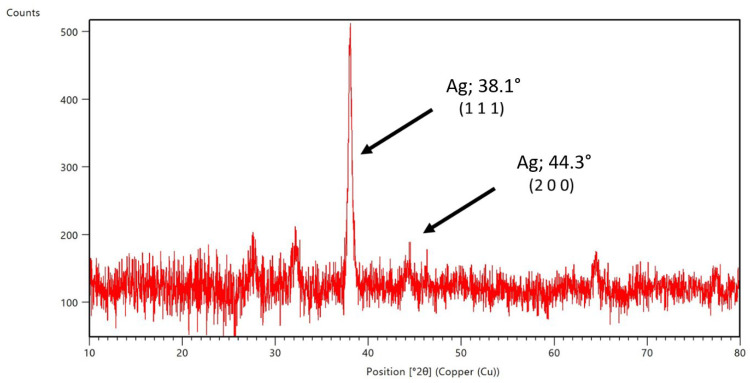
XRD diffractogram of Ma-AgNPs, background subtracted.

**Figure 5 pharmaceutics-16-01247-f005:**
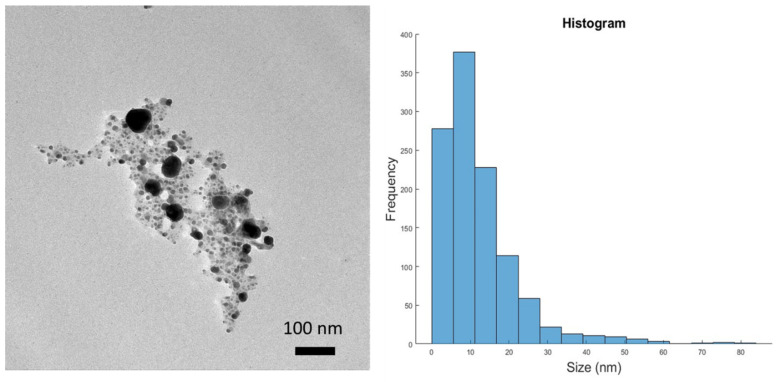
TEM microphotography of Ma-AgNPs and histogram distribution of particle size. Mean size 12.3 ± 10.0 nm.

**Figure 6 pharmaceutics-16-01247-f006:**
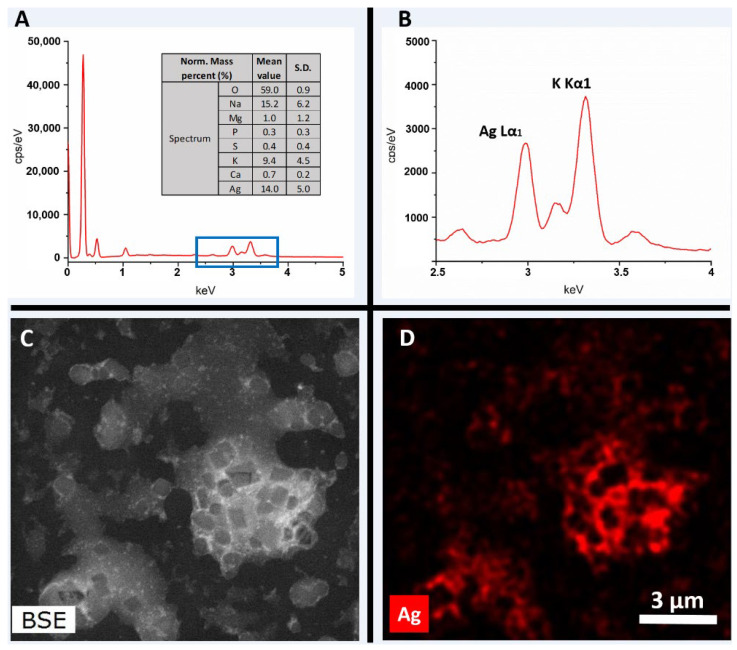
(**A**) EDS spectrum and elemental composition of Ma-AgNPs (S.D: Standard deviation; Norm. Mass percent: Normalized mass percent; cps: counts per second; keV: kilo electron volt). (**B**) Extended EDS spectra of Ma-AgNPs to confirm the presence of silver. (**C**) Backscattered Electron (BSE) image of Ma-AgNPs. (**D**) Elemental mapping of Ma-AgNPs.

**Figure 7 pharmaceutics-16-01247-f007:**
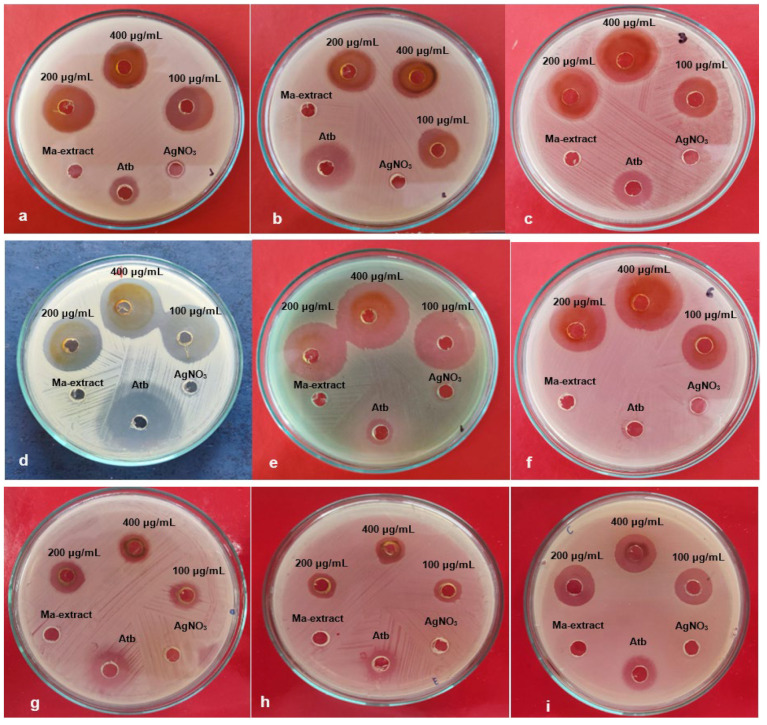
Antimicrobial activity of Ma-AgNPs prepared from Ma extract against (**a**) *B. cereus*; (**b**) *S. enterica* ATCC 14028; (**c**) *E. coli* ATCC 25922; (**d**) *S. aureus* ATCC 25923; (**e**) *P. aeruginosa* ATCC 27853; (**f**) *E. faecalis* ATCC 29212; (**g**) *K. pneumoniae* *; (**h**) *E. coli* *; and (**i**) *P. aeruginosa* *. * Multidrug-resistant bacteria.

**Figure 8 pharmaceutics-16-01247-f008:**
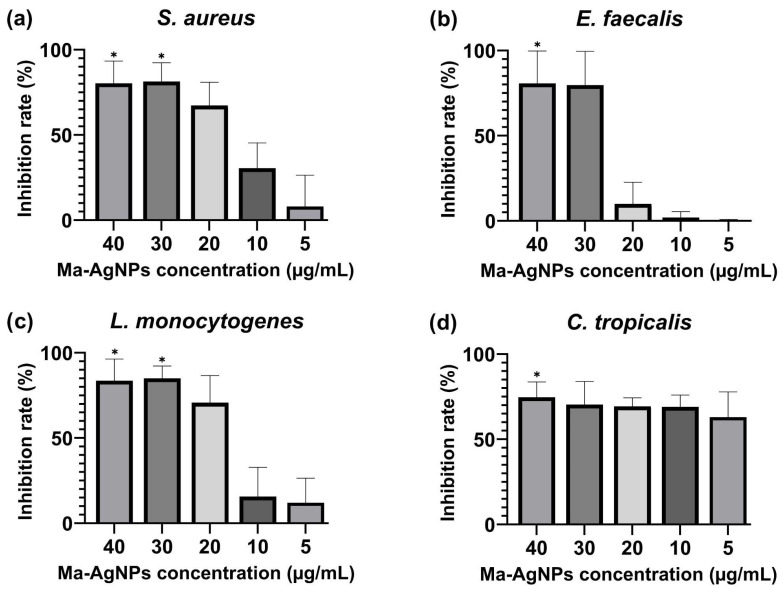
Inhibition of biofilm production by Ma-AgNPs. The graph shows the percentage of biofilm inhibition of four microorganisms: (**a**) *Staphylococcus aureus* ATCC 25923, (**b**) *Enterococcus faecalis* ATCC 29212, (**c**) *Listeria monocytogenes* ATCC 13932 and (**d**) *Candida tropicalis* ATCC 13803. Inhibition was tested after 24 h incubation with Ma-AgNPs at a 40–5 µg/mL concentration. Treatments at different concentrations were compared with 50% theoretical inhibition standard for the analysis of statistical significance using a two-way ANOVA test. All the values are mean ± SD, *p*-value (*) < 0.05.

**Figure 9 pharmaceutics-16-01247-f009:**
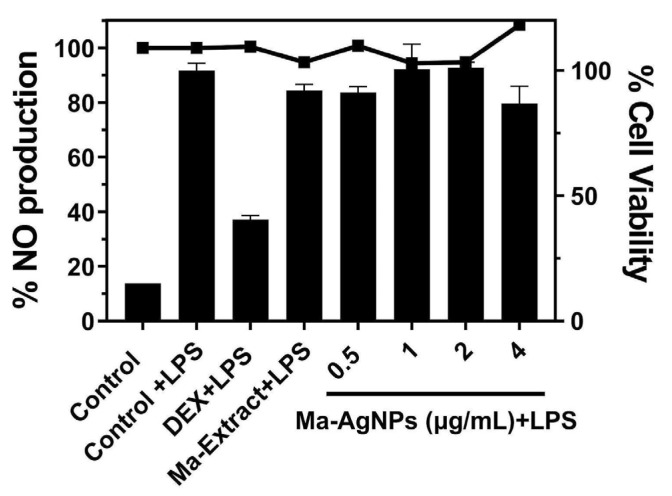
Anti-inflammatory activity of Ma-AgNPs. RAW264.7 cells were pretreated with the Ma-AgNPs and then stimulated with LPS. The percentage of NO production was calculated using cells treated with LPS only (control + LPS) as 100% NO production. Dexamethasone (DEX) was used as the positive control, and cells were incubated only with cell media as negative controls (control). Bars represent the percentage of NO production. Dots and lines represent the % cell viability for each corresponding data set.

**Table 1 pharmaceutics-16-01247-t001:** List of antibiotics and concentrations used as controls during the evaluation of antibacterial activity.

Bacterial Strain	Antibiotic
*B. cereus*	Chloramphenicol (20 µg/mL)
*S. enterica* ATCC 14028	Carbenicillin (100 µg/mL)
*E. coli* ATCC 25922	
*S. aureus* ATCC 25923	
*E. faecalis* ATCC 29212	
*P. aeruginosa* ATCC 27853	Tetracycline (10 µg/mL)
*K. pneumoniae* *	Nourseothricin (100 µg/mL)
*E. coli* *	
*P. aeruginosa* *	

* Multidrug-resistant bacteria.

**Table 2 pharmaceutics-16-01247-t002:** Individual phenolic compounds from *Mansoa alliacea*.

Phenolic Compound	Concentration (mg/100 g DW)
Gallic acid	19.3 ± 2.4
4-hydroxybenzoic acid	151.2 ± 1.1
Siringic acid	31.5 ± 3.1
*o*-Cumaric acid	182.5 ± 30.2
Luteolin	1903.2 ± 228.3
Luteolin isomer	870.8 ± 28.1
Apigenin	105.1 ± 9.1
Total phenolics	2840.3 ± 334.0

**Table 3 pharmaceutics-16-01247-t003:** Mean zone of inhibition (mm) and antimicrobial activity (MIC values in μg/mL) of Ma-AgNPs and antibiotics against multidrug and non-multidrug-resistant bacteria. The mean and standard deviation (SD) reported for each microbial strain at different concentrations were based on three biological replicates.

Bacterial Strain	Zone of Inhibition (mm)						MIC (µg/mL)
	Ma-AgNPs EA	Ma-AgNPs EB	Ma-AgNPs EC	Antibiotic	Ma Extract	AgNO_3_	
*B. cereus*	20.7 ± 0.5	19.7 ± 0.5	19.0 ± 0.0	14.3 ± 4.0 ^a^	NA	NA	2.5
*S. enterica* ATCC 14028	19.7 ± 2.1	18.0 ± 5.0	17.3 ± 6.1	12.0 ± 1.7 ^b^	NA	NA	1.3
*E. coli* ATCC 25922	20.7 ± 5.8	20.0 ± 4.0	15.3 ± 2.0	15.1 ± 3.0 ^b^	NA	NA	1.3
*S. aureus* ATCC 25923	31.3 ± 4.5	26.3 ± 3.2	25.9 ± 0.5	25.7 ± 2.5 ^b^	NA	NA	1.3
*E. faecalis* ATCC 29212	35.0 ± 4.2	22.5 ± 1.0	19.5 ± 1.0	13.0 ± 4.2 ^b^	NA	NA	5.0
*P. aeruginosa* ATCC 27853	25.7 ± 5.1	22.3 ± 3.5	21.0 ± 3.6	13.3 ± 3.2 ^c^	NA	NA	5.0
*K. pneumoniae* *	15.5 ± 1.0	12.0 ± 1.0	12.0 ± 1.0	15.0 ± 1.0 ^d^	NA	NA	5.0
*E. coli* *	11.7 ± 1.0	11.0 ± 1.0	12.0 ± 1.0	10.7 ± 1.0 ^d^	NA	NA	5.0
*P. aeruginosa* *	18.7 ± 0.5	17.3 ± 1.5	16.3 ± 0.5	12.7 ± 2.0 ^d^	NA	NA	10.0

* Multidrug-resistant bacteria; EA = Ma-AgNPs at 400 µg/mL; EB = Ma-AgNPS at 200 µg/mL; EC = Ma-AgNPs at 100 µg/mL. Antibiotics: a = chloramphenicol; b = carbenicillin; c = tetracycline; d = nourseothricin. NA = non-active at the tested concentrations.

**Table 4 pharmaceutics-16-01247-t004:** Antioxidant activity expressed as IC_50_ values for synthesized nanoparticles and plant extract after DPPH assay analysis. *p* values from the one-way ANOVA test are also presented.

Compound	IC_50_ (µg/mL)	*p* Values			
		Ma-AgNPs	AgNPs	Ma Extract	Ascorbic Acid
Ma-AgNPs	3.41 ± 0.89	-	0.9729	<0.0001	0.9630
AgNPs	5.54 ± 1.91		-	<0.0001	0.8203
Ma-extract	59.74 ± 13.40			-	<0.0001
Ascorbic acid	0.83 ± 0.16				-

**Table 5 pharmaceutics-16-01247-t005:** Therapeutic indexes (TI) and half maximal inhibitory concentration values (IC_50_) ^a^ of Ma-AgNPs and Ma extract against tumor and non-tumor cell lines at 72 h compared to cisplatin (CDDP). Values are expressed as mean ± standard deviation, n = 4.

Cells	Ma Extract	Ma-AgNPs		Cisplatin	
		IC_50_	TI ^b^	IC_50_	TI ^b^
MDA-MB-231	NA	5.4 ± 0.8	1.0	9.5 ± 2.3	0.4
SK-MEL-103	NA	4.4 ± 0.2	1.2	5.6 ± 0.8	0.7
HCT116	NA	5.2 ± 0.5	1.0	7.5 ± 0.5	0.5
HT29	NA	5.0 ± 0.8	1.1	2.9 ± 0.3	1.3
HeLa	NA	2.9 ± 0.3	1.9	2.3 ± 0.4	1.6
NIH3T3	NA	5.4 ± 1.4	---	3.7 ± 0.7	---

^a^ µg/mL; NA: not active. ^b^ IC_50_ (NIH3T3)/IC_50_ (tumor cell).

**Table 6 pharmaceutics-16-01247-t006:** Hemolytic activity of Ma-AgNPs.

	% Hemolysis	*p*-Value *
C−	0 ± 0.1	
C+	100.0 ± 3.0	
10 µg/mL	0 ± 0.4	9.85 × 10⁻¹⁷
20 µg/mL	0.5 ± 0.3	9.07 × 10⁻¹⁷
40 µg/mL	4.5 ± 3.6	6.22 × 10⁻²²

C−: negative control (PBS 1x), C+: positive control (10% Triton X-100). * The *p*-values were obtained using independent *t*-tests comparing the hemolytic activity of each Ma-AgNP (10 µg/mL, 20 µg/mL, and 40 µg/mL) against the positive control (C+).

## Data Availability

All tables were created by the authors. All sources of information were adequately referenced. There was no need to obtain copyright permission.

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
