# Peer review of "Phytosynthesis of Silver Nanoparticles Using Mansoa alliacea (Lam.) A.H. Gentry (Bignoniaceae) Leaf Extract: Characterization and Their Biological Activities"

_pharmaceutics, 2024, doi:10.3390/pharmaceutics16101247_

Round 1
Reviewer 1 Report
Comments and Suggestions for Authors
Comments:
It is an interesting work with relevant data but some corrections need to be made;
The title should correspond with the results.
1.- The summary should include the compounds or type of compounds found and mention which ones are responsible for the activity.
2.- In the introduction there is no clear evidence of traditional use. We found no relationship with the effect to be demonstrated.
3.- In the methodology, mention the place of collection, quantity of fresh plant and plant identification (include the number) and mention the sieve used for grinding.
4.- It is necessary to present the identification number of the species under study (Mansoa alliacea)
5. I find that very little plant is used (6 g of powder), and in addition there is talk of quantification and the extract is not taken to dryness (lyophilization) to measure the concentration used for evaluations and quantification of phenols and all the measurements that were made. It is necessary to have the performance of the data obtained is questionable.
6. A bipartition with ethyl acetate should be done to test where there is better reaction and activity. (Aqueous or organic fraction (ethyl acetate)
7.- Present positive and negative control in the biological data
8. There is no evidence of the compounds presented, the HPLC chromatogram of the extract must be included in the document indicating which of the compounds presented they correspond to). Add chromatograms in supplementary data.
9.- Present the registration number of the ethics or research committee.
10. The conclusions for further discussion. You have to be specific with your results. It has to be restructured.
Author Response
1.- The summary should include the compounds or type of compounds found and mention which ones are responsible for the activity.
We have revised the summary to explicitly mention the bioactive compounds responsible for the observed activities, such as polyphenols, flavonoids, anthocyanins, and tannins, which contribute to the enhanced biological properties of the synthesized nanoparticles.
2.- In the introduction there is no clear evidence of traditional use. We found no relationship with the effect to be demonstrated.
The introduction has been expanded to include more evidence on the traditional use of Mansoa alliacea, detailing its medicinal properties and applications in traditional healing practices in the Amazon region. Relevant references have been added to support these claims.
3.- In the methodology, mention the place of collection, quantity of fresh plant and plant identification (include the number), and mention the sieve used for grinding.
We appreciate the reviewer's comment. We have added a phrase indicating the precedence of vegetal material used. “One hundred healthy and fresh Mansoa alliacea leaves were purchased from the local market in the Amazon Region, Puyo city, Ecuador (1°29´3.671´´S, 77°59´39.172´´O). Regarding the plant identification, we do not have an identification number because the plant was not collected in its wild form; it is commercially available in the local market, and for this reason, it was not identified
Also, we have added the sieve used for grinding for clarity.
4.- It is necessary to present the identification number of the species under study (Mansoa alliacea)
We appreciate the reviewer's comment. Regarding the plant identification, we do not have an identification number because the plant was not collected in its wild form; it is commercially available in the local market, and for this reason, it was not identified.
- I find that very little plant is used (6 g of powder), and in addition there is talk of quantification and the extract is not taken to dryness (lyophilization) to measure the concentration used for evaluations and quantification of phenols and all the measurements that were made. It is necessary to have the performance of the data obtained is questionable.
We appreciate the reviewer's comment. The extraction was carried out with a 6:100 (w/v) ratio of the plant with aqueous solvent. Then, the extract was dried in a freeze-dryer and the dry extract was weighed with which the different analyses and biological activities were carried out. This statement has been included in the section 2.2
- A bipartition with ethyl acetate should be done to test where there is better reaction and activity. (Aqueous or organic fraction (ethyl acetate)
We chose to synthesize nanoparticles using a green method that involves only water for the extraction process, avoiding the use of polluting solvents and emphasizing the importance of environmentally friendly synthesis.
7.- Present positive and negative control in the biological data
We acknowledge the reviewer's suggestion. We have included appropriate positive and negative controls for each biological assay (Antibiogram, MIC, Anti-inflammatory, Anti-tumor, and Hemolytic Assay) to validate the results, and this information has been incorporated into the revised manuscript.
- There is no evidence of the compounds presented, the HPLC chromatogram of the extract must be included in the document indicating which of the compounds presented they correspond to). Add chromatograms in supplementary data.
We appreciate the reviewer's comment. We have added the respective chromatogram section 3.1/Figure 1
9.- Present the registration number of the ethics or research committee.
Thank you for your valuable feedback. The cell lines utilized in this study were sourced from reputable suppliers, specifically ATCC (American Type Culture Collection) and Applied Biological Materials Inc. (abm, Richmond, BC, Canada). These cell lines are commercially available and widely recognized for their reliability, eliminating the need for additional institutional ethical approval. We have included the specific numbers for the cell lines in the Materials and Methods section (2.8. Cell culture experiments) to ensure traceability.
- The conclusions for further discussion. You have to be specific with your results. It has to be restructured.
The conclusions have been refined to provide a more concise summary of the key findings and their potential implications in biomedical applications.
Reviewer 2 Report
Comments and Suggestions for Authors
In the MS pharmaceutics-3182201 the authors performed an interesting, complex, and extensive study on Ma-AgNPs, including preparation, phytochemical analysis, and pharmacological properties.
The following comments are available below:
I. Title
Please put the entire scientific name in the title: Mansoa alliacea (Lam.) A.H.Gentry (Bignoniaceae).
II. Introduction
Lines 65-66: The transition from AgNPs to the studied plant species (M. alliacea) is too steep. Please briefly present some general data from the scientific literature regarding the biogenic synthesis of AgNPs using plant extracts.
Line 66: Please include in the MS text the entire scientific name and family of Mansoa alliacea (Lam.) A.H. Gentry (Bignoniaceae). Then, more data about habitat, traditional therapeutic uses, etc., should be provided.
Line 81 - Please explain the abbreviation SGD.
Line 86 - Please put the Abbreviation in parentheses (Ma-AgNPs).
III. Materials and Methods:
As an overview, this section is blurred and very agglomerated; all methods are unclearly described. Please put all very detailed aspects in the supplementary material and maintain essential data in the MS text: reagents, working conditions, sample and control preparation for each analysis, concentrations used, and suitable references for each method.
1. The reviewer suggests introducing a first subsection named "Materials," which would include all reagents, chemicals, antibiotics, culture media, defibrinated sheep blood, and all cell lines used in the present study and provide their provenance (Company name, City, Country). Thus, all techniques should be presented more briefly. Please simplify the text as much as possible. More details should be included in the supplementary material.
2. Subsection 2.1. could be entitled briefly, "Synthesis of silver nanoparticles using Mansoa alliacea leaf extract," and all steps are included: plant harvesting, extract preparation, and nanoparticle biosynthesis.
Line 93: Please indicate the place and the mode of harvesting, the period of the year, and the voucher specimen number.
3. Subsection 2.2. The method used for total polyphenol content evaluation is the most known. Please present it in a shorter manner.
4. Subsection 2.3. HPLC
This section is difficult to read as two agglomerated paragraphs. Please separate it into subsections according to the essential steps: Standard Solutions preparation, sample preparation, working conditions, and results interpretation.
Lines 136-144: Please put the calibration curves in the supplementary material.
5. Subsection 2.5. Antibacterial activity of Ma-AgNPs
This subsection is very agglomerated. The reviewer suggests separating it into a few sub-subsections describing the main steps: bacterial inoculum, samples and controls solutions preparation, lab technique, results interpretation, etc.
Line 165: Please show the preparation of bacterial inoculum.
Lines 166-170: Please check and reformulate the phrase from these lines. B. cereus and K. pneumoniae do not have an ATCC number. Moreover, do the MDR strains E coli and P. aeruginosa differ from those previously mentioned? If the response is positive, please indicate their provenance.
Table 1 (line 188) contains 5 bacterial species: Listeria sp has not previously been mentioned, and other species (Salmonella and Klebsiella sp) are missing. Please check and correct.
6. Subsection 2.6. Antibiofilm activity
The same general comments are available.
Please show the reference used for this method.
Line 192. Please mention the entire name of the fungal species and indicate the provenance of clinical isolates.
Usually, the antibacterial/antifungal and antibiofilm activity is tested on the same bacterial/fungal species, thus ensuring the study uniformity. The authors are encouraged to proceed accordingly. If not, they should include a strong motivation for their selection in the discussion section.
7. The methodology for DPPH free radical scavenging activity is so well-known; please present it briefly and show the samples and controls.
8. Subsection 2.8. Please provide a suitable reference for the method used, include the reference to the mentioned protocol (line 245), and briefly present the Griess reagent system. Moreover, please show the samples and controls (negative and positive).
9. Subsection 2.9. Cytotoxic activity is better because the authors used only an MTT assay on tumor and normal cell lines.
Please organize this subsection into a few sub-subsections (as the reviewer previously suggested) and identify the samples and controls (negative and/or positive). Please include suitable references for the method used.
10. Line 283. p<0.05 is enough.
V. Results:
Please respect the same order of analyses as in Material and Methods. Moreover, verify that all data from the results are mentioned in Materials and Methods. For example, Ma extract and AgNO3 (antibacterial activity), Cisplatin (cytotoxic activity).
1. Subsection 3.1. could be entitled "Bioactive constituents" because phenolic compounds were identified and quantified.
Table 2. Please revise the Table caption and avoid "average values."
2. Please put Figure 1 in Supplementary Material.
3. Table 3. Please revise the Table 3 caption, put the correspondent concentrations (not dilutions) for Ma-AgNPs, indicate the values obtained for controls, and show the conventional antibiotic for each bacterial sp. in the table footer. The MIC values should be recorded similarly (Mean+/- SD). Moreover, calculate the p-values and show the statistically different ones (p<0.05).
Figure 7: why AgNO3?
4. Please include in the supplementary material the table with antibiofilm activity values expressed as a mean +/-SD and clearly show the p-values.
Figure 8 - please indicate what values are compared to establish the p-value.
5. Subsection 3.9. Please use the same notations in the MS text and Table 4: AgNPs, Ma-AgNPs, Ma-extract, and show p-values.
6. Subsection 3.10. please show the p-values in Tables 5 and 6.
7. Subsection 3.11. Please register all values obtained (mean +/- SD)+ p-values in a table and present them in the MS text; put Figure 10 in the supplementary material.
8. Subsection 3.12. Please revise Table 7, evidencing the samples and controls. In the current form, without explanations in the table footer, it is difficult for the reader to understand them. Please show the p-values.
VI. Abstract, Discussion and Conclusions
The authors are invited to revise these sections after all previously mentioned aspects were solved.
Author Response
- Title|. Please put the entire scientific name in the title: Mansoa alliacea (Lam.) A.H.Gentry (Bignoniaceae).
We appreciate your insightful comments. We added (Lam.) A.H.Gentry (Bignoniaceae) in the title.
- Introduction: Lines 65-66: The transition from AgNPs to the studied plant species (M. alliacea) is too steep. Please briefly present some general data from the scientific literature regarding the biogenic synthesis of AgNPs using plant extracts.
We have modified the text to create a smoother transition between the discussion of silver nanoparticles (AgNPs) and the specific use of Mansoa alliacea for nanoparticle synthesis, providing a clearer context for our choice of this plant species. Supporting literature was added to provide clearer context.
- Line 66: Please include in the MS text the entire scientific name and family of Mansoa alliacea (Lam.) A.H. Gentry (Bignoniaceae). Then, more data about habitat, traditional therapeutic uses, etc., should be provided.
We have included in the title the entire scientific name and more data in the introductory section supporting its traditional use.
- Line 81 - Please explain the abbreviation SGD.
The abbreviation "SGD" has been clarified within the text to ensure understanding.
- Line 86 - Please put the Abbreviation in parentheses (Ma-AgNPs).
We have consistently used the abbreviation "Ma-AgNPs" throughout the manuscript for "Mansoa alliacea-derived Silver Nanoparticles."
III. Materials and Methods:
As an overview, this section is blurred and very agglomerated; all methods are unclearly described. Please put all very detailed aspects in the supplementary material and maintain essential data in the MS text: reagents, working conditions, sample and control preparation for each analysis, concentrations used, and suitable references for each method.
- The reviewer suggests introducing a first subsection named "Materials," which would include all reagents, chemicals, antibiotics, culture media, defibrinated sheep blood, and all cell lines used in the present study and provide their provenance (Company name, City, Country). Thus, all techniques should be presented more briefly. Please simplify the text as much as possible. More details should be included in the supplementary material.
We thank the reviewer for his insightful comments. We have added a section called: “2.1. Reagents and Standards”. We have also simplified the text to present it more briefly.
- Subsection 2.1. could be entitled briefly, "Synthesis of silver nanoparticles using Mansoa alliacea leaf extract," and all steps are included: plant harvesting, extract preparation, and nanoparticle biosynthesis.
We appreciate your insightful comments. We have changed the title of the subsection.
- Line 93: Please indicate the place and the mode of harvesting, the period of the year, and the voucher specimen number.
We appreciate the reviewer's comment. The mode of harvesting do not apply because the plant was purchased at a local market in the Amazon Region, Puyo city, Ecuador (1°29´3.671´´S, 77°59´39.172´´O) . Regarding the voucher specimen number we do not have an identification number because the plant was not collected in its wild form; it is commercially available in the local market, and for this reason, it was not identified. The period of year was included in section 2.3.
- Subsection 2.2. The method used for total polyphenol content evaluation is the most known. Please present it in a shorter manner.
The method to evaluate total polyphenols has been shortened as suggested by the reviewer.
- Subsection 2.3. HPLC
This section is difficult to read as two agglomerated paragraphs. Please separate it into subsections according to the essential steps: Standard Solutions preparation, sample preparation, working conditions, and results interpretation.
We appreciate the reviewer's comment. The subsection 2.2. Quantification of phenols has been modified.
- Lines 136-144: Please put the calibration curves in the supplementary material.
Thank you for the reviewer's comment. We have provided the requested information in a separate file, which will be uploaded as non-published material.
- Subsection 2.5. Antibacterial activity of Ma-AgNPs
This subsection is very agglomerated. The reviewer suggests separating it into a few sub-subsections describing the main steps: bacterial inoculum, samples and controls solutions preparation, lab technique, results interpretation, etc.
We appreciate your insightful comments. We have added subsections according to the method used and have reduced the main text.
- Line 165: Please show the preparation of bacterial inoculum.
We appreciate your comments. We have added the preparation of bacterial inoculum for both techniques in the corresponding section.
- Lines 166-170: Please check and reformulate the phrase from these lines. B. cereus and K. pneumoniae do not have an ATCC number. Moreover, do the MDR strains E coli and P. aeruginosa differ from those previously mentioned? If the response is positive, please indicate their provenance.
We appreciate your insightful comments. We have corrected the text according to the reviewer suggestions. The MDR are different strains, we have decided to add a paragraph about the provenance of the strains used in this study. The new subsection is “2.5.1. Bacterial strains”
- Table 1 (line 188) contains 5 bacterial species: Listeria sp has not previously been mentioned, and other species (Salmonella and Klebsiella sp) are missing. Please check and correct.
We appreciate the reviewer's comment. We have reviewed this mistake and now is corrected.
- Subsection 2.6. Antibiofilm activity
The same general comments are available. Please show the reference used for this method.
We thank the reviewer for the suggestion. The text has been revised to clarify some of the confusing data.
- Line 192. Please mention the entire name of the fungal species and indicate the provenance of clinical isolates. Usually, the antibacterial/antifungal and antibiofilm activity is tested on the same bacterial/fungal species, thus ensuring the study uniformity. The authors are encouraged to proceed accordingly. If not, they should include a strong motivation for their selection in the discussion section. Carlos
Thank you for your insightful observation, the description of the fungal species has been clarified as all strains used in the biofilm inhibition assay during this study belong to the ATCC microorganism collection. The correct ATCC code has been clarified.
Additionally, it is important to mention that only the microorganisms with strong biofilm-forming capabilities, as reported previously in literature, were used for the anti-biofilm evaluation in our study. Therefore, we have listed all the strains used in a new section “2.2. Bacterial and fungal strains”, only 4 of those strains were selected for the anti-biofilm evaluation as mentioned due to their previous usage in literature for this type of essay. We are deeply thankful for this observation and valuable feedback, we believe the new changes have strengthened our manuscript.
- The methodology for DPPH free radical scavenging activity is so well-known; please present it briefly and show the samples and controls.
We thank the reviewer for the suggestion. This section has been clarified as requested. Additionally, the method is described as brief as possible due to the modifications made from the literature to adapt it to a 96 microtiter plate.
- Subsection 2.8. Please provide a suitable reference for the method used, include the reference to the mentioned protocol (line 245), and briefly present the Griess reagent system. Moreover, please show the samples and controls (negative and positive).
Thank you for your insightful suggestions. We have updated this section to include a suitable reference for the method used and have briefly described the Griess reagent system to enhance clarity. Additionally, we have specified the samples and controls utilized in the experiment, including both negative and positive controls for both the assay and the experiment.
- Subsection 2.9. Cytotoxic activity is better because the authors used only an MTT assay on tumor and normal cell lines.
Please organize this subsection into a few sub-subsections (as the reviewer previously suggested) and identify the samples and controls (negative and/or positive). Please include suitable references for the method used.
Thank you for your valuable feedback. The MTT assay was chosen over a general cell viability assay to assess antitumor activity because it specifically measures cellular metabolic activity, which is a direct indicator of cell proliferation, making it particularly valuable for evaluating the cytostatic and cytotoxic effects of antitumor agents. In contrast, general cell viability assays, such as those measuring membrane integrity, may not be as sensitive in detecting subtle changes in cell proliferation, especially when the cells' growth is inhibited without causing immediate cell death. This approach is supported by numerous studies who have validated the MTT assay's effectiveness in assessing the impact of anticancer drugs on cell proliferation (doi: 10.1016/j.toxlet.2005.07.001, doi:10.1016/j.acthis.2018.02.005, doi: 10.3390/diseases6040085).
Additionally, we have clearly identified the samples and controls (both negative and positive) used in the MTT assay, and we have included suitable references for the method employed. Lastly, we have reorganized section 2.8, 2.9, and 2.10 into sub-subsections to improve clarity. The new structure includes:
2.8 Cell Culture Experiments; 2.8.1 Cell Lines; 2.8.2 Anti-inflammatory Activity; 2.8.3 Anti-tumor Activity.
- Line 283. p<0.05 is enough.
Thank you for your valuable feedback. The description of the p-value has been rewritten.
- Results:
Please respect the same order of analyses as in Material and Methods. Moreover, verify that all data from the results are mentioned in Materials and Methods. For example, Ma extract and AgNO3 (antibacterial activity), Cisplatin (cytotoxic activity).
- Subsection 3.1. could be entitled "Bioactive constituents" because phenolic compounds were identified and quantified.
We appreciate the reviewer's comment. We have merged sections 2.2 and 2.3 under a more general title, "Quantification of Phenols." We chose not to use "bioactive compound" as it refers to a broader category encompassing thousands of molecules. Additionally, we have revised the title of subsection 3.1 to "3.1. Identification of Bioactive Constituents" (line 297).
- Table 2. Please revise the Table caption and avoid "average values."
We appreciate the reviewer´s comment. We changed the title to “Individual phenolic compounds from Mansoa alliacea”
- Please put Figure 1 in Supplementary Material.
Thank you for your comment, we have placed the figure in the supplementary section.
- Table 3. Please revise the Table 3 caption, put the correspondent concentrations (not dilutions) for Ma-AgNPs, indicate the values obtained for controls, and show the conventional antibiotic for each bacterial sp. in the table footer. The MIC values should be recorded similarly (Mean+/- SD). Moreover, calculate the p-values and show the statistically different ones (p<0.05).
We thank the reviewer for the suggestion. We have added the correspondent concentrations for Ma-AgNPs. Additionally, we have added two columns for controls which are non active, and we added the antibiotics used for each bacteria in the table footer. As explained in the methods, the MIC is determined using fixed concentrations, therefore a standard deviation cannot be obtained and p-values ​​cannot be calculated either.
- Figure 7: why AgNO3?
In response to the reviewer's question about the use of AgNO₃ in Figure 7, AgNO₃ was used as a precursor for the synthesis of Ma-AgNPs.
- Please include in the supplementary material the table with antibiofilm activity values expressed as a mean +/-SD and clearly show the p-values.
We thank the reviewer for the suggestion. The new table with all the new data has been modified as supplementary material
- Figure 8 - please indicate what values are compared to establish the p-value.
We appreciate this observation too and the description has been changed to indicate that the comparison test was done towards a 50% inhibition standard
- Subsection 3.9. Please use the same notations in the MS text and Table 4: AgNPs, Ma-AgNPs, Ma-extract, and show p-values.
We thank the reviewer for the suggestion. The names in the table have been modified and p-values have been added.
- Subsection 3.10. please show the p-values in Tables 5 and 6.
Thank you for your comments. For Table 5 (IC50 values), we included standard deviations to show variability but did not perform statistical comparisons between samples. The table is intended to describe the IC50 values, with the standard deviation reflecting measurement precision. For Table 6 (Therapeutic Index values), p-values cannot be calculated because the therapeutic index is a ratio of the IC50 values of the antitumor agents to those of normal cells and it represents a measure of selectivity rather than a statistical measure. We hope this clarifies the data presentation.
- Subsection 3.11. Please register all values obtained (mean +/- SD)+ p-values in a table and present them in the MS text; put Figure 10 in the supplementary material.
Thank you for your suggestion. We created a table that registered all values obtained (mean ± SD) and included it in the supplementary material (Supp Material Table B) along with Figure 10. The mean values were used to calculate the IC50 values (Table 5). The manuscript was updated accordingly.
- Subsection 3.12. Please revise Table 7, evidencing the samples and controls. In the current form, without explanations in the table footer, it is difficult for the reader to understand them. Please show the p-values.
Thank you for your observation. In the modified version of the manuscript we included a foot table explaining the positive and negative controls included in the table, and the t-student test comparing the hemolytic activity of each nanoparticle concentration. In addition we added a column to the table showing the p-values obtained from the t-student test.
- Abstract, Discussion and Conclusions
The authors are invited to revise these sections after all previously mentioned aspects were solved.
Reviewer 3 Report
Comments and Suggestions for Authors
Revision Comments:
The keyword should be revised. Anti-tumor should be added, and nanoparticles should be written 1 time.
The authors have not provided the data about plant’s identification.
Which method was used for the preparation of extract? Support it with literature.
In line 117, 300 seconds?
How wells were prepared and what was the size of wells?
The authors have not provided ATCC numbers for cell lines.
“2.2. Determination of total polyphenol contents” where are results for this activity. I suggest that authors should read all the MS carefully before submission.
What standard antibiotic was used in antibacterial assay?
Discussion should be improved.
The conclusion is too long. It should be concise.
Comments on the Quality of English LanguageMinor editing of English language required
Author Response
- The keyword should be revised. Anti-tumor should be added, and nanoparticles should be written 1 time.
Thank you for your suggestion. Changes were made in the keyword section.
- The authors have not provided the data about plant’s identification.
Regarding the plant identification, we do not have an identification number because the plant was not collected in its wild form; it is commercially available in the local market, and for this reason, it was not identified.
- Which method was used for the preparation of extract? Support it with literature.
We appreciate the reviewer's comment. We have added a reference for preparation of extract.
- In line 117, 300 seconds?
We thank the reviewer for the observation. The error has been corrected for clarity.
- How wells were prepared and what was the size of wells?
Thank you for your comment. The size, brand and specific preparation of the wells have been indicated in the materials and methods section.
- The authors have not provided ATCC numbers for cell lines.
Thank you for your feedback. We have included the reference numbers for each cell line in the Materials and Methods section (2.8.1 Cell lines).
- “2.2. Determination of total polyphenol contents” where are results for this activity. I suggest that authors should read all the MS carefully before submission.
The results are found in section 3.1.
- What standard antibiotic was used in antibacterial assay?
Thank you for your comment. The standard antibiotics used in antibacterial assay were specified in the Table 1.
- Discussion should be improved.
The discussion was thoroughly reviewed and modified where appropriate.
- The conclusion is too long. It should be concise.
The conclusions have been refined to provide a more concise summary of the key findings and their potential implications in biomedical applications.
Reviewer 4 Report
Comments and Suggestions for Authors
The paper is on a relevant topic, interesting, comprehensive and well written. However, there are a few comments.
1. Minor drawbacks include the poor quality of Figures 2 and 3.
2. Also, the composition of the particles in Figure 6 is unclear. If these are plant matter residues, why is there sodium, potassium and calcium, but no nitrogen and carbon?
3. As far as I understand, the authors used different concentrations of nanoparticles in different experiments. Why? Is there an optimal concentration of nanoparticles for most of the applications studied?
4. Please discuss the related works:
Naganthran A, Verasoundarapandian G, Khalid FE, Masarudin MJ, Zulkharnain A, Nawawi NM, Karim M, Che Abdullah CA, Ahmad SA. Synthesis, Characterization and Biomedical Application of Silver Nanoparticles. Materials (Basel). 2022 Jan 6;15(2):427. doi: 10.3390/ma15020427.
Prasad A. S. (2016). Iron oxide nanoparticles synthesized by controlled bio-precipitation using leaf extract of Garlic Vine (Mansoa alliacea). Mat. Sci. Semicond. process. 53, 79–83. 10.1016/J.MSSP.2016.06.009.
Author Response
- Minor drawbacks include the poor quality of Figures 2 and 3.
Thank you for your comment. The figure 2 and 3 has been replaced using a major quality.
- Also, the composition of the particles in Figure 6 is unclear. If these are plant matter residues, why is there sodium, potassium and calcium, but no nitrogen and carbon?
Dear Reviewer, thank you for your comment. We excluded carbon from the measurements as the samples were collocated on a carbon double tape; in such a case, carbon is the dominant element to be measured. There is also no interest in measuring carbon, as it is evident that it is part of the plant composition. Moreover, carbon detection is quite not precise as only translating the sample into the atmosphere will recover it of a small layer of carbon, inducing an increase from 5% to 20% in carbon content (see for example excellent reference book on SEM of Goldstein et al.). Nitrogen is detected below a reasonable limit (for EDS technic), so we do not include it.
- As far as I understand, the authors used different concentrations of nanoparticles in different experiments. Why? Is there an optimal concentration of nanoparticles for most of the applications studied?
Thank you for your comment. The different concentrations of nanoparticles were used to evaluate their effects across a range of conditions and to determine the concentration-dependent response for each specific application. The optimal concentration may vary depending on the biological assay and the type of cells or microorganisms being tested, which is why multiple concentrations were assessed.
- Please discuss the related works:
Naganthran A, Verasoundarapandian G, Khalid FE, Masarudin MJ, Zulkharnain A, Nawawi NM, Karim M, Che Abdullah CA, Ahmad SA. Synthesis, Characterization and Biomedical Application of Silver Nanoparticles. Materials (Basel). 2022 Jan 6;15(2):427. doi: 10.3390/ma15020427.
Prasad A. S. (2016). Iron oxide nanoparticles synthesized by controlled bio-precipitation using leaf extract of Garlic Vine (Mansoa alliacea). Mat. Sci. Semicond. process. 53, 79–83. 10.1016/J.MSSP.2016.06.009.
We thank the reviewer for these suggestions. Both references have been reviewed and properly cited in the manuscript.
Round 2
Reviewer 1 Report
Comments and Suggestions for Authors
Comments:
I have reviewed the corrections made and the following is necessary:
The identification of the plant is of utmost importance. You cannot believe that it is the one you are studying, even if it is bought in the market. You must first make sure that Mansoa alliacea is the plant you are studying. These data, even if they are new, cannot be published without being completely sure that it is the species being studied. If someone wants or wishes to repeat the experiment, it must be shown that it is the plant. You cannot not include it; a taxonomist must identify it and save a species for future reference.
In figure 1, it is not specified what equipment was used.
Author Response
Dear Reviewer,
Thank you for your valuable comments regarding the identification of the plant Mansoa alliacea. We would like to clarify that following your suggestion, we have identified M. alliacea by a specialized taxonomist. The specimen is deposited in the Herbarium QUPS-Ecuador with the identification code 4431 and is available for future reference. Additionally, an image of the plant is included in the supplementary material (Figure A) for further verification.
We appreciate your attention to this critical aspect of our study.
Reviewer 2 Report
Comments and Suggestions for Authors
The reviewer appreciates the authors' efforts to revise the MS according to the previous review report. They suitably responded to all comments and suggestions.
Author Response
We thank the reviewer for their invaluable comments, which have helped enrich our work.
Reviewer 3 Report
Comments and Suggestions for Authors
The comments have been addressed properly by authors. Now the manuscript is improved.
Author Response

(The authors gave the same response as above.)
